# Coincident postsynaptic activity gates presynaptic dopamine release to induce plasticity in *Drosophila* mushroom bodies

**Kohei Ueno\*, Ema Suzuki, Shintaro Naganos, Kyoko Ofusa, Junjiro Horiuchi, Minoru Saitoe\***

Learning and Memory Project, Tokyo Metropolitan Institute of Medical Science, Setagaya, Japan

**Abstract** Simultaneous stimulation of the antennal lobes (ALs) and the ascending fibers of the ventral nerve cord (AFV), two sensory inputs to the mushroom bodies (MBs), induces long-term enhancement (LTE) of subsequent AL-evoked MB responses. LTE induction requires activation of at least three signaling pathways to the MBs, mediated by nicotinic acetylcholine receptors (nAChRs), NMDA receptors (NRs), and D1 dopamine receptors (D1Rs). Here, we demonstrate that inputs from the AL are transmitted to the MBs through nAChRs, and inputs from the AFV are transmitted by NRs. Dopamine signaling occurs downstream of both nAChR and NR activation, and requires simultaneous stimulation of both pathways. Dopamine release requires the activity of the rutabaga adenylyl cyclase in postsynaptic MB neurons, and release is restricted to MB neurons that receive coincident stimulation. Our results indicate that postsynaptic activity can gate presynaptic dopamine release to regulate plasticity.

**\*For correspondence:** ueno-kh@ igakuken.or.jp (KU); saito-mn@ igakuken.or.jp (MS)

**Competing interests:** The authors declare that no competing interests exist.

## Introduction

Dopamine (DA) signaling is required for associative learning and synaptic plasticity (*Jay, 2003*; *Kim et al., 2007*; *Puig and Miller, 2012*; *Qin et al., 2012*). However, its precise function in these processes is not known. In associative learning, where a conditioned stimulus or CS (e.g. a neutral odor) is paired with an unconditioned stimulus or US (e.g. aversive electrical shocks or appetitive food rewards), DA activity often correlates with US presentation (*Aso et al., 2010*; *Burke et al., 2012*; *Riemensperger et al., 2005*; *Schwaerzel et al., 2003*; *Waddell, 2013*). Indeed, in *Drosophila*, the activation of different subsets of dopaminergic neurons (DANs) can substitute for the presentation of aversive or appetitive stimuli during training (*Aso et al., 2010*; *Burke et al., 2012*; *Claridge-Chang et al., 2009*; *Schwaerzel et al., 2003*; *Waddell, 2013*). This suggests that DA functions to transmit US information to olfactory association areas in the *Drosophila* brain. However, results from mammals suggest that the function of DA is more complex. While the DAN activity initially correlates with US presentation, as animals learn to associate the CS with the US, activity becomes correlated with the CS (*Wise, 2004*). DAN activity also correlates with the unexpectedness of the US presentation. In reward or punishment prediction error paradigms, a reward or punishment that is expected causes little change in activity, while a reward or punishment that is obtained or withheld unexpectedly induces strong increases or decreases in activity (*Bromberg-Martin et al., 2010*; *Busto et al., 2010*; *Schultz, 2013*; *Waddell, 2013*). To more precisely understand the role of DA in learning and plasticity, here we examined DA release properties in dissected *Drosophila* brains.

In *Drosophila* aversive olfactory conditioning, presentation of a relatively neutral odor is paired with presentation of aversive electric shocks (*Quinn et al., 1974*; *Tully and Quinn, 1985*). Flies learn

to associate the odor with the aversive stimulus and subsequently avoid the odor. This association forms in the mushroom bodies (MBs) of the *Drosophila* brain (*Heisenberg, 2003*). Odor information is transmitted to the MBs from the antennal lobes (ALs) via projection neurons (*Ito et al., 1998*; *Stocker, 1994*), while shock information is likely transmitted from the body through the ascending fibers of the ventral nerve cord (AFV). Previously, we and others developed a functional imaging system using isolated *Drosophila* brains to study synaptic plasticity in the MBs (*Ueno et al., 2013*; *Wang et al., 2008*). Electrical stimulation of the ALs or AFV induces $Ca^{2+}$ influx in the MBs of dissected brains. However, simultaneous stimulation of both the ALs and AFV causes long-term enhancement (LTE) of subsequent AL-induced $Ca^{2+}$ responses in the MBs that lasts for at least two hours.

While LTE is a physiological phenomenon, it shares several similarities with associative learning. Both LTE and learning require activity of at least three neurotransmitter receptors, nicotinic acetylcholine receptors (nAChRs), NMDA receptors (NRs), and D1Rs (*Kim et al., 2007*; *Qin et al., 2012*; *Ueno et al., 2013*; *Xia et al., 2005*). In addition, LTE formation requires associativity between the two input stimuli, since increasing the stimulus intensity or duration of one input is insufficient to bypass the requirement of the other (*Ueno et al., 2013*). LTE also displays input specificity. MB responses to AL stimulation, but not AFV stimulation, are enhanced. Similarly, during associative learning, responses to odors are modified, but not responses to electrical shocks (*Quinn et al., 1974*). Finally, LTE can be extinguished by repetitive AL stimulation in the absence of AFV stimulation (*Ueno et al., 2013*), a phenomenon reminiscent of learning extinction (*Quinn et al., 1974*).

The dissected brain imaging system allows characterization of synaptic plasticity using a combination of mutant and pharmacological analyses. In this study, we examined when DA is released during LTE induction using both the fluorescent $Ca^{2+}$ reporter, G-CaMP (*Nakai et al., 2001*) or R-GECO1 (*Zhao et al., 2011*), and the vesicular exocytosis probe, synapto-pHluorin (spH) (*Miesenböck et al., 1998*). We determined that stimulation of either the ALs or AFV alone is unable to evoke release. Instead, simultaneous activation of postsynaptic MBs, by both AL and NR-mediated AFV inputs, is required. Application of DA to the MBs on its own is sufficient to induce LTE, and DA release requires *rutabaga* adenylyl cyclase (Rut-AC) in the MBs. We propose that sensory information is conveyed to the MBs through acetylcholine and glutamate, and that coincident stimulation of postsynaptic neurons is required to gate presynaptic DA release to induce plasticity in specific targets.

## Results

### AFV stimulation causes vesicular release from glutamatergic terminals

Simultaneous stimulation of the AL and AFV (AL + AFV) induces LTE that can be measured at the tip of MB vertical lobes. (*Ueno et al., 2013*) (*Figure 1A*). To first verify whether cholinergic PNs, which connect the ALs to the MBs, transmit AL stimuli to the MBs, we expressed spH in PNs to visualize synaptic vesicle exocytosis (*Figure 1B*). We also expressed the red-fluorescent $Ca^{2+}$ indicator, R-GECO1 (*Zhao et al., 2011*) in MBs to verify that AL stimulation induces $Ca^{2+}$ responses in the MBs (*Figure 1B*). In the absence of AL stimulation, we observed baseline spH fluorescence in PN terminals in the MB calyx (short arrowheads), lateral horn (long arrowheads), and in PN dendrites in the AL. spH fluorescence was also observed in the MBs, since the GH146-PN driver also labels anterior posterior lateral (APL) neurons which innervate the MBs (*Tanaka et al., 2008*). Upon stimulation of the ALs, we observed increased spH fluorescence in PN terminals, including those in the MB calyx, the MB region receiving inputs from PNs. We also observed robust $Ca^{2+}$ responses in the vertical lobes of the MBs (*Figure 1C*), indicating that AL stimulation causes both synaptic vesicle exocytosis from PN terminals on the MB calyx and $Ca^{2+}$ response in MB lobes.

Next, to test whether DA signaling transmits AFV stimuli to the MBs, we expressed spH in (tyrosine hydroxylase) TH-expressing DA neurons and R-GECO1 in the MBs (*Figure 1D*). In this case, baseline spH fluorescence was observed in DAN axonal terminals in various brain regions, including the MB vertical lobes (dotted region in *Figure 1D*). When we next stimulated the AFV, we again observed robust $Ca^{2+}$ responses in the MBs. However, we did not observe increased spH fluorescence in the MB vertical lobes (*Figure 1E*). This suggests that, although DA signaling is required for LTE, TH-DA neurons may not be directly responsible for transmitting AFV information to the MBs.

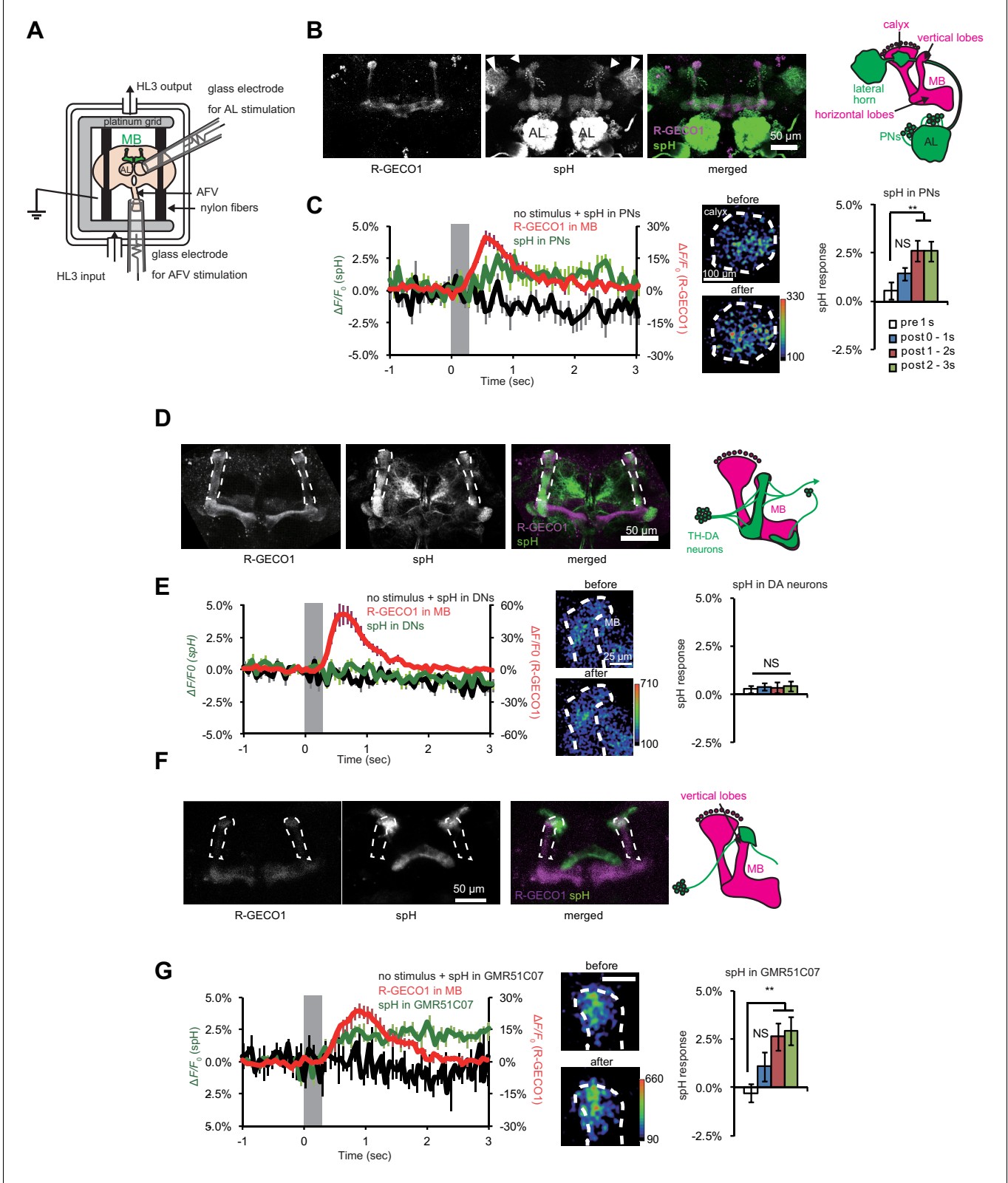

**Figure 1.** Synaptic vesicle exocytosis upon stimulation of the ALs and AFV. (**A**) Schematic diagram of the stimulation/recording system. (**B**) Fluorescent images of a representative *MB-LexA,LexAop-R-GECO1/UAS-spH, GH146-GAL4* brain. (**C**) AL stimulation induced Ca²⁺ responses in the MBs, and evoked vesicle exocytosis from PN terminals of the PNs. Left panel, the time course of spH and R-GECO1 fluorescence changes upon AL stimulation. spH fluorescence changes upon AL stimulation (green) and background (black) were measured in the MB calyces (typical images are shown in middle

*Figure 1 continued*

panels), and R-GECO1 fluorescence (red) was measured at the terminals of the MB vertical lobes. The gray bar in the right panel indicates AL stimulation. Right panel, spH responses were obtained by subtracting background fluorescence changes from stimulation-evoked fluorescence changes. One-way ANOVA indicates significant differences in fluorescence ($F_{3,40}$ = 4.769, p=0.006). N = 11. (D) Fluorescent images of a *MB-LexA, LexAop-R-GECO1/UAS-spH;TH-GAL4/+* brain. (E) AFV stimulation did not evoke exocytosis from TH-DA neuron terminals. Left panel, time course of spH and R-GECO1 fluorescence changes upon AFV stimulation. Middle panels, typical images of spH fluorescence changes at TH-DA neuron terminals on the vertical MB lobes. Right panel, one-way ANOVA indicates no significant differences in spH fluorescence ($F_{3,36}$ = 0.061, p=0.980). N = 10. (F) Fluorescent images of a *MB-LexA,LexAop-R-GECO1/UAS-spH;GMR51C07-GAL4/+* brain. (G) AFV stimulation evoked vesicle exocytosis at glutamatergic terminals at the tip of vertical MB lobes. Changes in spH and R-GECO1 fluorescence produced upon AFV stimulation. Right panel, One-way ANOVA indicates significant differences in spH flurorescence ($F_{3,20}$ = 4.927, p=0.010). N = 6.

The following figure supplement is available for figure 1:

**Figure supplement 1.** Activity of glutamatergic neurons is required for normal STM.

We previously showed that the NR antagonist, MK801, impairs AFV-evoked MB $Ca^{2+}$ responses and LTE (*Ueno et al., 2013*), suggesting that glutamate (Glu), rather than DA, may transmit AFV signals to the MBs. Furthermore, synaptic terminals immunoreactive for the vesicular Glu transporter, VGLUT, have been identified at the tip of the MB vertical lobes, where LTE is strongly induced (*Daniels et al., 2008*). Thus, to test whether AFV stimulation induces vesicular exocytosis from glutamatergic terminals, we expressed spH using the *GMR51C07-VGlut-GAL4* driver, and R-GECO1 in MBs using *MB-LexA*. *GMR51C07-VGlut-GAL4* contains approximately 3 KB of the 5′ flanking sequence of the *VGlut* gene and drives GAL4 expression in a subset of glutamatergic neurons innervating the tip of the MB vertical lobes (*Figure 1F*). In the absence of AFV stimulation, we observed baseline spH fluorescence at the tips of the MB vertical lobes and in the central complex. AFV stimulation caused significant increases in both spH and R-GECO1 fluorescence at the MB vertical lobe tips (*Figure 1G*), suggesting that stimulation induces exocytosis from *GMR51C07-VGlut* terminals onto the MBs.

If glutamatergic input from GMR51C07 neurons transmits somatosensory information from the AFV to the MBs, inhibiting activity of GMR51C07 neurons should impair learning or memory. To test this, we next inhibited synaptic output from GMR51C07 neurons during learning, by expressing *shi*[ts], a temperature-sensitive dominant-negative variant of dynamin (*Kitamoto, 2001*) in these cells. When *GMR51C07>shi*[ts] flies were trained in an aversive olfactory associative paradigm at the permissive temperature, short-term memory (STM) was unaffected compared to controls. In contrast, when these flies were trained at the restrictive temperature, memory was significantly impaired (*Figure 1—figure supplement 1*). Thus, GMR51C07 cells both release glutamate to the MBs upon AFV stimulation, and are required during formation of aversive memories.

Consistent with idea that Glu transmits AFV inputs required for LTE to the MBs, we found that a *Drosophila* NR1 hypomorphic mutant, *Nmdar1*[EP331] (*Xia et al., 2005*), is defective for both AFV-induced MB $Ca^{2+}$ responses (*Figure 2A*), and LTE (*Figure 2B*). Furthermore, we found that AFV stimulation could be replaced by the direct application of NMDA by micropipette onto the terminals of the MB vertical lobes (*Figure 2C*), to induce LTE (*Figure 2D*). Application of NMDA in the absence of AL stimulation did not induce LTE, indicating that NMDA can replace AFV, but not AL, stimulation. To determine whether NRs in the MBs mediate NMDA signaling, we next knocked down NRs in the MBs by expressing inverted repeats of *Nmdar2* (*Nmdar2-IR*) from the *OK107-GAL4* driver, and found that coupled AL stimulation and NMDA application (AL+ NMDA) failed to induce LTE (*Figure 2E*). Knocking down NRs in the MBs also inhibited learning (3 min memory) in adult *Drosophila*, again suggesting a link between LTE and learning-associated plasticity (*Figure 2—figure supplement 1*). Taken together, our results suggest that AFV input to the MBs is transmitted via Glu neurons onto MB NRs.

## D1R activation is sufficient to induce LTE

If ACh and Glu transmit AL and AFV inputs to the MBs, what is the function of DA? *Drosophila* D1Rs, encoded by the *DopR* gene, are required for LTE induced by AL + AFV stimulation (*Ueno et al., 2013*). To determine the epistatatic relationship between DA and Glu signaling during

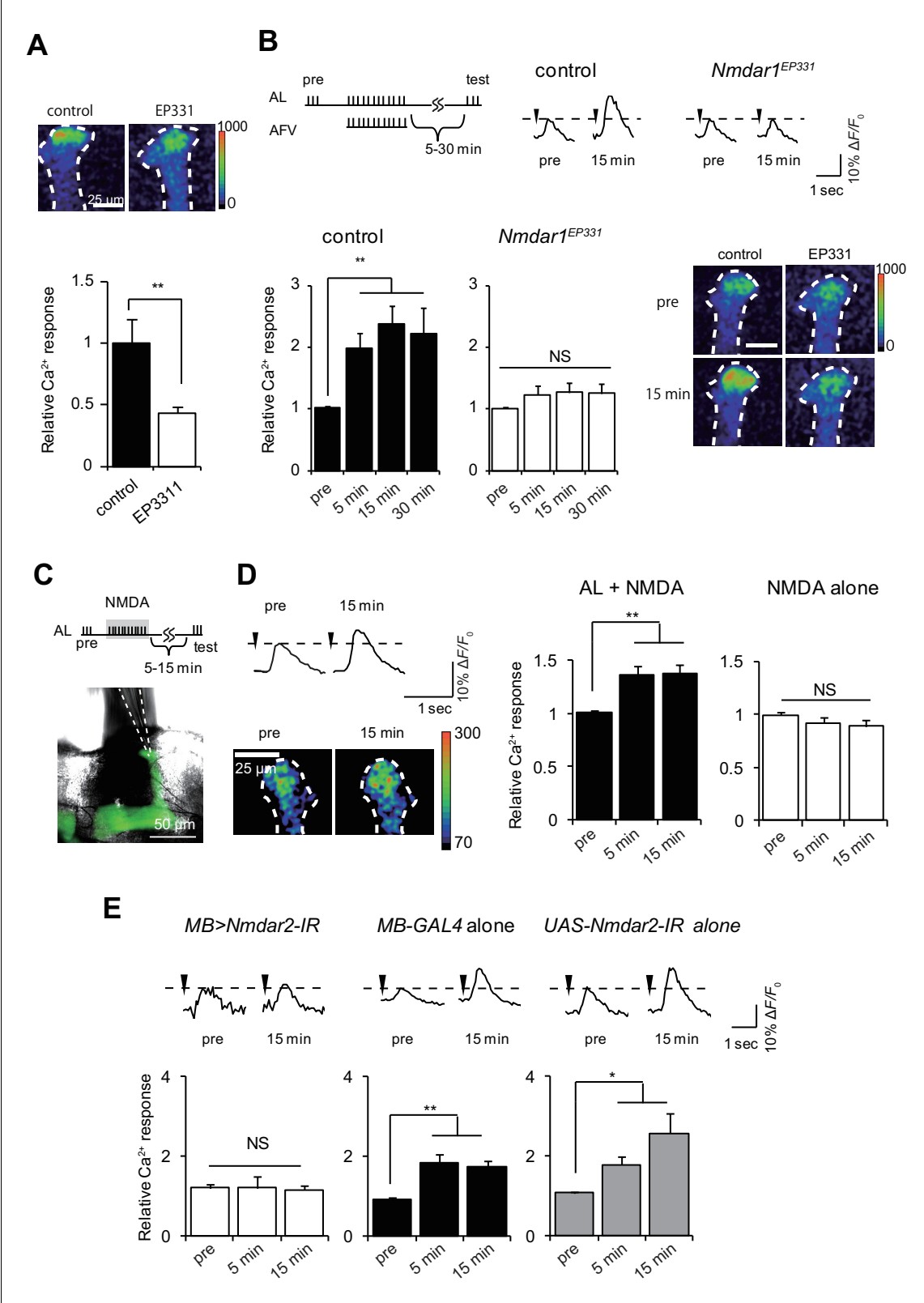

**Figure 2.** NR activation can substitute for AFV stimulation during LTE induction. (**A**) AFV-evoked Ca$^{2+}$ responses in the MBs are significantly diminished in *Nmdar1*$^{EP331}$ mutants. Upper panel, typical images of G-CaMP fluorescence immediately after AFV stimulation in *MB-LexA, LexAop-G-CaMP2* (control) and M*B-LexA, LexAop-G-CaMP2; Nmdar1*$^{EP331}$ mutants. (**B**) LTE induced by AL+AFV stimulation is abolished in *Nmdar1* mutants. The upper panels show a schematic diagram of the LTE induction protocol, and typical traces of AL-induced Ca$^{2+}$ responses before and 15 min after induction.
*Figure 2 continued on next page*

Figure 2 continued

Arrowheads indicate AL-stimulation. Lower panels, one-way ANOVA indicates significant LTE in controls (filled columns, $F_{3,20}$ = 9.347, p<0.001) but not in *Nmdar1* mutants (blank columns, $F_{3,20}$ = 0.621, p=2.900). N = 6. (C) Schematic of LTE induction by simultaneous AL stimulation and puff application of NMDA (AL + NMDA). Dotted lines in the lower panel indicate the position of a micropipette. (D) LTE induced by simultaneous AL stimulation and NMDA application. One-way ANOVA indicates significant LTE upon AL + NMDA ($F_{2,18}$ = 6.062, p=0.010, N = 7) but not upon NMDA puff alone ($F_{2,18}$ = 1.470, p=0.256, N = 7). (E) Expressing *Nmdar2-IR* in the MBs from the *OK107-GAL4* driver (*MB-LexA, LexAop-G-CaMP2/+; UAS-Nmdar2-IR/+; OK107-GAL4/+*) abolishes LTE. Left panel, one-way ANOVA did not identify significant differences between time points ($F_{2,15}$ = 0.031, p=0.969, N = 6). One-way ANOVA indicates significant LTE in control flies containing the *OK107-GAL4* driver alone ($F_{2,12}$ = 5.169, p=0.024 for *GAL4* alone, N = 5), or the *UAS-Nmdar2-IR* construct alone ($F_{2,15}$ = 11.388, p<0.001, N = 6).

The following figure supplement is available for figure 2:

**Figure supplement 1.** NRs in the MBs are required for normal learning.

LTE induction, we examined AL + NMDA-induced LTE, and found that it is abolished in *DopR*[f02676], *DopR* null mutant, brains (*Kim et al., 2007*) (*Figure 3A*). Furthermore, addition of 50 µM of the *Drosophila* D1R antagonist, butaclamol (*Sugamori et al., 1995*) prevented induction of AL + NMDA-induced LTE in wild-type brains (*Figure 3B*). These results indicate that D1R functions downstream of Glu/NR signaling during LTE formation. Thus, we next performed bypass experiments to determine whether application of DA could replace NMDA application during LTE induction. When AL stimulation was paired with application of DA (AL + DA), we observed significant LTE (*Figure 3C*). Surprisingly, we also detected LTE formation upon application of DA alone in the absence of AL stimulation (*Figure 3D*). Although this result was unexpected since it seems to bypass the specificity requirement for formation of associations, it is consistent with a model where DA signaling functions after association of simultaneous AL and AFV inputs during LTE formation. Further supporting this model, DA-induced LTE was not impaired in brains from *Nmdar1*[EP331] mutants (*Figure 3E*).

We previously showed that LTE induced by simultaneous AL + AFV stimulation requires the expression of D1Rs specifically in the MBs (*Ueno et al., 2013*). To determine whether DA-induced LTE also depends on MB expression of these receptors, we carried out rescue experiments using an inducible *DopR* mutant. The *DopR*[f02676] allele consists of a UAS-containing transposon inserted into the *DopR* locus, and *DopR*[f02676] mutants do not express *DopR*, unless they also contain a GAL4 driver (*Kim et al., 2007*; *Thibault et al., 2004*). The DA-induced LTE was fully restored in *DopR*[f02676] mutants when *DopR* was expressed from *c747-Gal4*, a MB driver predominantly expressing GAL4 in $\alpha/\beta$ and $\gamma$ lobes (*Aso et al., 2009*) (*Figure 3F*). Dose response curves plotting LTE magnitude as a function of DA concentration were not different when DA was applied alone or with simultaneous AL stimulation (*Figure 3G*), suggesting that the DA-induced LTE is unlikely to be an artifact of excess DA application.

To determine whether DA functions during LTE formation, or during maintenance or expression, we induced LTE by DA application, and added the D1R antagonist, SCH23390 (*Sugamori et al., 1995*), either during DA application, or after washout during maintenance or expression phases. The addition of antagonist during DA treatment abolished LTE (*Figure 4A*), while addition during maintenance and expression phases had no effect (*Figure 4B*). This discounts the possibility that residual DA remaining after washout is a responsible expression or maintenance of LTE, and instead indicates that DA is required during LTE induction. DA application induced LTE even under external $Ca^{2+}$ free conditions, and in the presence of tetrodotoxin (TTX), which prevents the generation of action potentials (*Figure 4C*), suggesting that DA release and MB D1R activation occur at a late step of LTE induction where subsequent neurotransmission is not required.

## Coincidently activated MB neurons gate DA release to induce LTE

If DA signaling occurs downstream of AL + NMDA stimulation, DA release may be regulated by coincident AL and AFV inputs. To test this, we re-examined vesicular exocytosis from TH-DA neuron terminals by expressing spH in TH-DA neurons, and observed significant elevation of spH fluorescence from TH-DA neuron terminals on the MB vertical lobes upon simultaneous AL stimulation and NMDA application (*Figure 5A*). spH fluorescence was unchanged upon either AL stimulation or NMDA application alone. Likewise, spH fluorescence increased significantly in TH-DA neuron

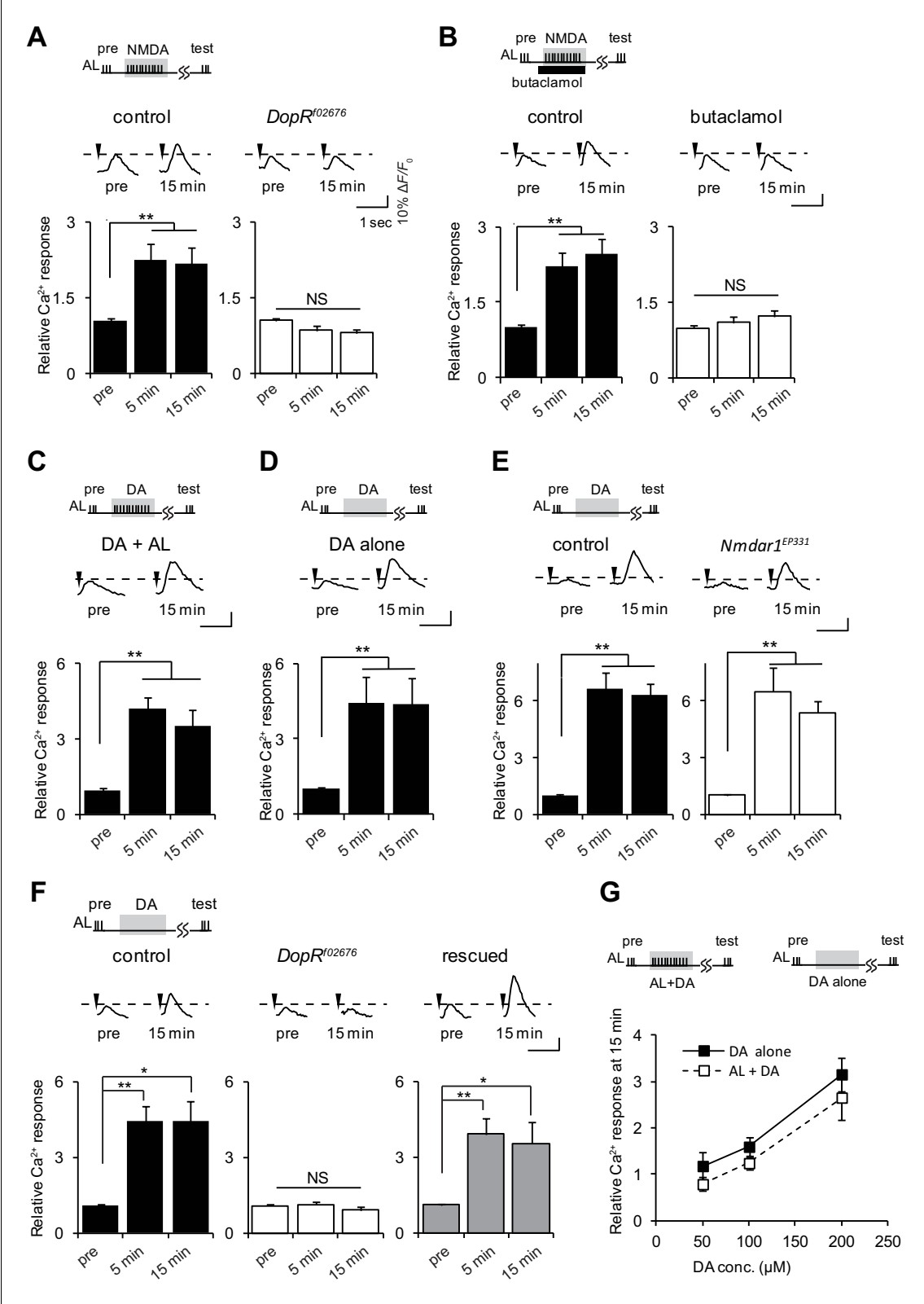

**Figure 3.** D1R activation is necessary and sufficient to induce LTE. (**A**) *DopR*[f02676] mutations suppress LTE induced by AL + NMDA stimulation. Left panel, *MB-LexA, LexAop-G-CaMP2/+*, one-way ANOVA ($F_{2,18}$ = 5.932, p<0.001). Right panel, *MB-LexA, LexAop-G-CaMP2/+; DopR*[f02676], one-way ANOVA ($F_{2,18}$ = 3.479, p=0.053). N = 7 for both genotypes. (**B**) LTE induced by AL + NMDA stimulation in *MB-LexA, LexAop-G-CaMP2* brains is abolished by the addition of the D1R antagonist 10 min prior to AL + NMDA stimulation. Control, $F_{2,15}$ = 22.148, p<0.001, butaclamol, $F_{2,15}$ = 0.334, *Figure 3 continued on next page*

Figure 3 continued

p=0.801, N = 6 for all data. (C) LTE is induced by simultaneous AL + DA stimulation in *c309-GAL4; UAS-G-CaMP* brains, $F_{2,18}$ = 5.374, p=0.015, N = 7. (D) LTE can be induced by application of DA alone. One-way ANOVA ($F_{2,18}$ = 25.306, p<0.001, N = 7). (E) DA-induced LTE in *Nmdar1*[EP331] mutants is indistinguishable from DA-induced LTE in controls. Left panel, *MB-LexA, LexAop-G-CaMP2/+*, one-way ANOVA ($F_{2,18}$ = 22.591, p<0.001, N = 7). Right panel, *MB-LexA, LexAop-G-CaMP2/+; Nmdar1*[EP331], one-way ANOVA ($F_{2,15}$ = 10.335, p=0.002, N = 6). (F) DA-induced LTE is restored in *DopR*[f02676] mutants by expressing a *DopR* transgene in the MBs. Control (*MB-LexA, LexAop-G-CaMP2/+*), one-way ANOVA ($F_{2,15}$ = 8.261, p=0.004). *DopR*[f02676] (*MB-LexA, LexAop-G-CaMP2/+; DopR*[f02676]), one-way ANOVA ($F_{2,15}$ = 0.111, p=0.351). Rescue (*MB-LexA, LexAop-G-CaMP2/c747-GAL4; DopR*[f02676]), one-way ANOVA ($F_{2,15}$ = 5.556, p=0.016). N = 6. (G) LTE induced by DA alone is indistinguishable from LTE induced by AL + DA stimulation. Peak LTE responses plotted as a function of DA concentration. Two-way ANOVA demonstrated significant differences in Ca$^{2+}$ responses due to DA concentration ($F_{2,30}$ = 22.49, p<0.0001), but no differences due to the stimulation protocol (DA alone versus DA + AL, $F_{1,30}$ = 2.869, p=0.1007, N = 6).

terminals upon simultaneous AL and AFV stimulation (***Figure 5B***). These results are consistent with a model where presynaptic DA release is gated by postsynaptic MB neurons that are activated by coincident AL and AFV inputs. Alternately, it is possible that while cholinergic and glutamatergic inputs to the MBs are required for LTE, separate cholinergic and glutamatergic inputs to DA neurons

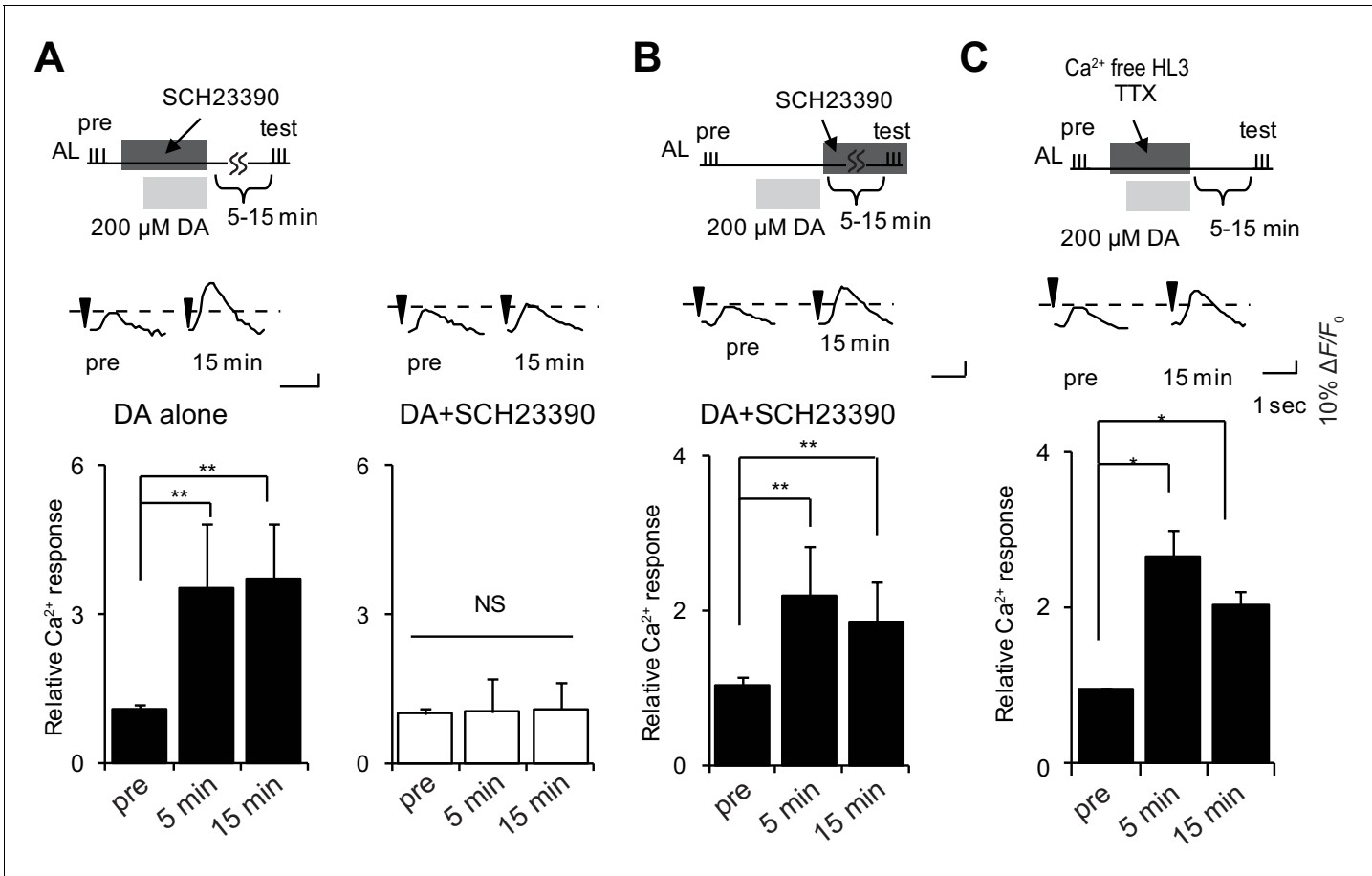

**Figure 4.** DA is required for induction but not maintenance or expression of LTE. (A) DA-induced LTE was abolished by applying 100 µM SCH23390 10 min prior to treatment with 200 µM DA in *c309-GAL4; UAS-G-CaMP* brains. One-way ANOVA indicates significant differences between time points in DA alone ($F_{2,15}$ = 11.050, p=0.001) but not in DA + SCH23390 ($F_{2,15}$ = 0.051, p=0.950). N = 6. (B) LTE was unaffected when SCH23390 was applied immediately after DA treatment in *c309-GAL4; UAS-G-CaMP* brains. One-way ANOVA indicates significant differences between time points ($F_{2,15}$ = 14.745, p=<0.001, N = 6). (C) DA-induced LTE was not suppressed in external Ca$^{2+}$ free solution containing 1 mM EGTA and 100 nM tetrodotoxin (TTX) in *c309-GAL4; UAS-G-CaMP* brains. One-way ANOVA indicates significant differences between time points in control ($F_{2,15}$ = 9.926, p=0.002, N = 6).

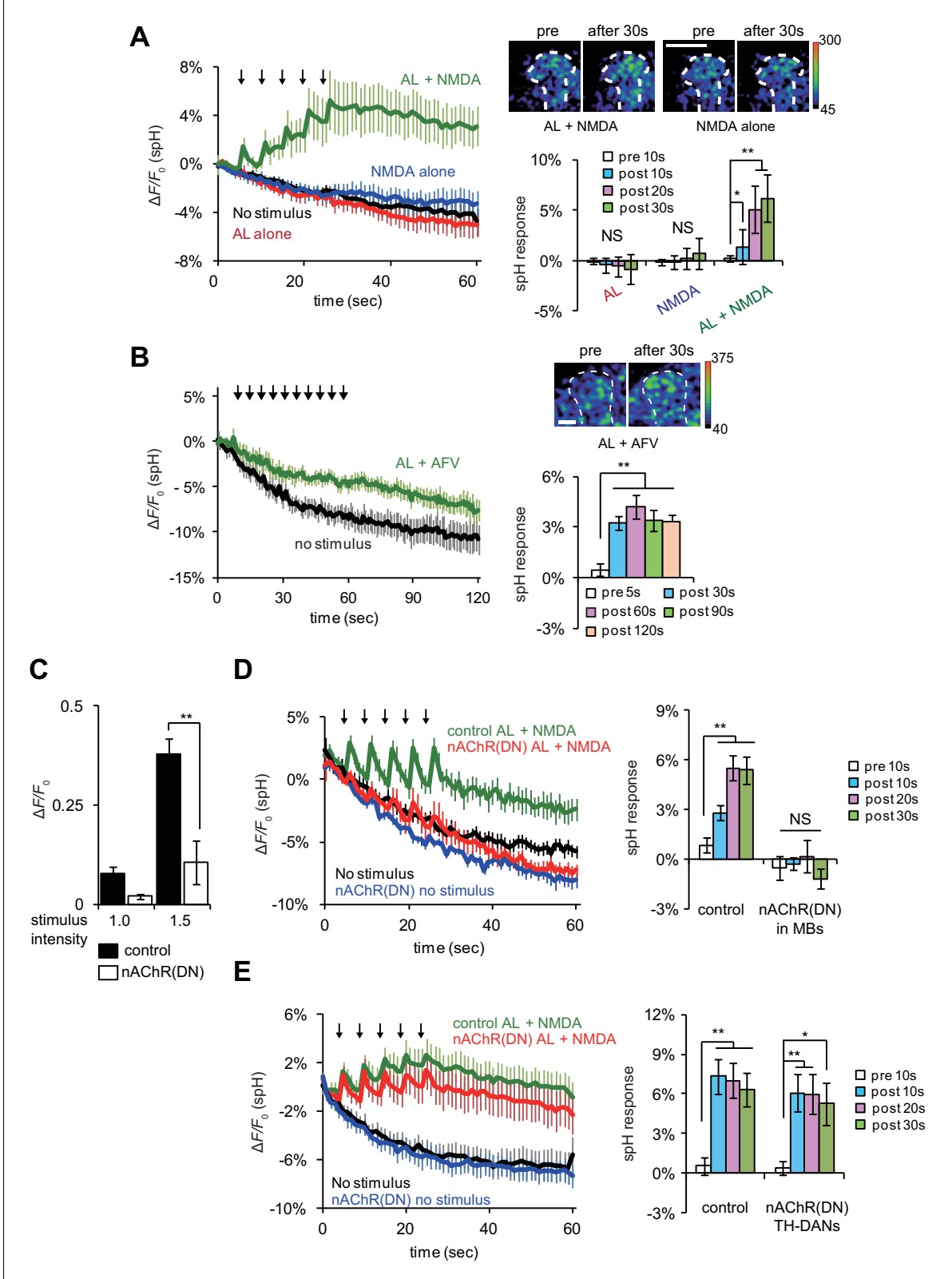

**Figure 5.** Coincident AL stimulation and NMDA application induces DA release. (**A**) Time course of spH fluorescence changes and quantification of spH responses at TH-DA terminals on the vertical lobes of the MBs upon indicated treatments. Brains were first exposed to 100 μM NMDA for 1 min while spH fluorescence was scanned (blue traces). Following washout and a 3 min interval, spH fluorescence was scanned while brains were subjected to 5 trains of AL stimulation (red traces). After another 3 min interval, spH fluorescence was scanned while brains were exposed simultaneously to

*Figure 5 continued on next page*

*Figure 5 continued*

NMDA and AL stimulation (green traces). Black traces show spH fluorescence in the absence of stimulation. Arrows represent AL stimulation. One-way ANOVA indicates significant differences between time points upon AL + NMDA stimulation ($F_{4,25}$ = 6.593, p=0.001, N = 6). (B) Time course of spH fluorescence changes and quantification of spH responses at TH-DA terminals induced by AL + AFV stimulation. One-way ANOVA indicates significant differences between time points ($F_{4,25}$ = 6.593, p=0.001, N = 6). (C) $Ca^{2+}$ responses in the MBs upon AL stimulation are inhibited by expressing dominant negative nAChRs, nAChR(DN)s, in the MBs. Control, *UAS-G-CaMP3/+;;OK107* and nAChR(DN), *UAS-G-CaMP3/UAS- nAChRα7$^{Y195T}$;;OK107*. Stimulus intensities of 1.0 and 1.5 refer to pulse durations of 1.0 and 1.5 ms, respectively. Two-way ANOVA indicates significant differences due to stimulus intensity ($F_{1,20}$ = 20.58, p=0.0002) and genotype ($F_{1,20}$ = 28.7, p<0.0001). N = 6 for all genotypes. (D) Expression of nAChR(DN)s in MB neurons abolishes AL + NMDA stimulation-induced DA release. One-way ANOVA indicates significant differences between time points upon AL + NMDA stimulation in control ($F_{3,32}$ = 11.888, p<0.001) but not nAChR(DN) brains ($F_{3,32}$ = 0.664, p=0.058). (E) Expression of nAChR(DN)s in TH-DA neurons does not affect the DA release induced by AL + NMDA stimulation. Control, *UAS-spH/+; TH-GAL4/+* and nAChR DN, *UAS-spH/ UAS-nAChRα7$^{Y195T}$; TH-GAL4*. One-way ANOVA indicates significant differences between time points upon AL + NMDA stimulation in control ($F_{3,28}$ = 9.59, p=0.0002) and nAChR DN brains ($F_{3,28}$ = 3.45, p=0.0298). N = 8 for all genotypes.

may also be required for DA release. To test this possibility, we used a dominant negative form of the nAChRα7 subunit, *nAChRα7$^{Y195T}$*, to inhibit nicotinic AChR activation (*Mejia et al., 2013*). We confirmed that the expression of *nAChRα7$^{Y195T}$* in MBs effectively disrupts both cholinergic transmission from AL to MBs and release of DA from TH-DA terminals upon AL + NMDA stimulation (*Figure 5C and D*). However, when we expressed *nAChRα7$^{Y195T}$* in TH-DA neurons, we still observed significant increases in spH fluorescence from TH-DA neuron terminals upon coincident AL + NMDA stimulation (*Figure 5E*). This suggests that coincident inputs to MBs, but not TH-DA neurons, are required for DA release.

We next examined the mechanism through which MB neurons might detect coincident inputs. In *Drosophila,* the *rutabaga* (*rut*) gene encodes an adenylyl cyclase (Rut-AC) that is required in the MBs during associative learning (*Busto et al., 2010*), and is proposed to function as a coincidence detector. We found that Rut-AC is also required for LTE formation (*Figure 6A*). To determine whether Rut-AC is required for DA release following coincident AL + NMDA stimulation, we examined release in *rut$^{2080}$* mutants (*Levin et al., 1992*), and observed significant impairments (*Figure 6B and D*). Significantly release was restored when a *rut* transgene (*rut$^+$*) was re-expressed in the MBs from the *30Y-GAL4* MB driver (*Figure 6C and D*), indicating that the presynaptic DA vesicle release is regulated by post-synaptic MB neurons through a Rut-AC-dependent mechanism.

Previous work has shown that most TH-DA neurons project their terminals bilaterally onto the MBs in both hemispheres of the brain (*Mao and Davis, 2009*). Thus, if DA release is controlled strictly by anterograde, action potential-dependent mechanisms, DA should be released onto the MBs on both hemispheres. On the other hand, our findings suggest that the DA release and induction of plasticity will be restricted to specific locations where post-synaptic MB neurons receive coincident stimulation. The ALs are bilateral structures, and anatomical and physiological studies show that PNs connect the ALs only to ipsilateral MBs (*Marin et al., 2002*; *Ueno et al., 2013*; *Wong et al., 2002*). Thus, we next tested whether simultaneous AL + NMDA stimulation induces DA release bilaterally, or only unilaterally to the MB ipsilateral to AL stimulation. We monitored vesicular exocytosis from TH-DA neuronal terminals in response to simultaneous AL + NMDA stimulation (*Figure 7A*), and observed DA release only on the MB lobes ipsilateral to AL stimulation (*Figure 7B and C*). Thus, postsynaptic gating mechanisms are able to restrict the location of DA release, even though DA neurons innervate extensive brain areas.

## Discussion

DA has been proposed to function either to modulate behaviors based on physiological states, including differing motivational or hunger states, or to more directly transmit associative input, during associative learning. In this study, we demonstrate that for LTE in the MBs, DA is unlikely to transmit sensory information, but instead, plays a critical mechanistic role in LTE establishment at a step after sensory input. DA is not released by activation of individual sensory inputs, but instead requires coincident stimulation, indicating that DA functions downstream of a coincidence detector. Blocking DA signaling abolishes LTE, while DA release alone is sufficient to induce LTE, demonstrating the essential nature of DA in LTE formation. Although LTE is a phenomenon observed *in vitro,*

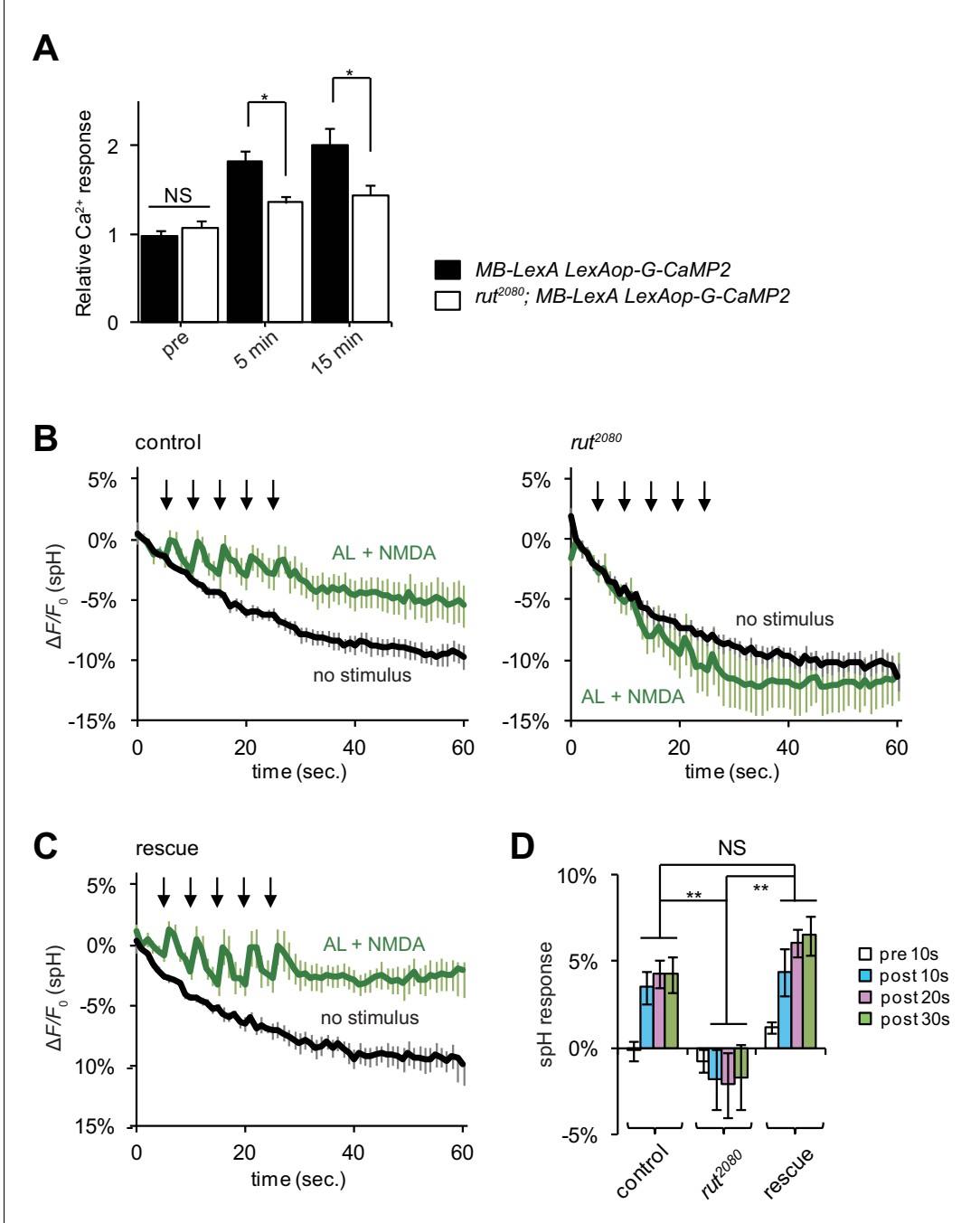

**Figure 6.** LTE and DA release require Rut-Ac activity in the MBs. (**A**) LTE induced by simultaneous AL + AFV stimulation is suppressed in *rut* mutants. N = 7 for both control (*MB-LexA, LexAop-G-CaMP2*) and *rut* mutants (*rut²⁰⁸⁰; MB-LexA, LexAop-G-CaMP2*). (**B**) DA release induced by AL + NMDA stimulation is abolished in *rut* mutants. Time course of spH fluorescent changes at TH-DA terminals in controls (*LexAop-spH/+; TH-LexA/+*), and *rut²⁰⁸⁰* mutants (*rut²⁰⁸⁰; LexAop-spH/UAS-rut; TH-LexA/+*) upon AL + NMDA stimulation. N = 8 for all genotypes. (**C**) MB expression of *rut⁺* restores DA release upon AL + NMDA stimulation. The time course of spH fluorescent changes and spH responses at TH-DA terminals in *rut²⁰⁸⁰* mutants expressing a *rut⁺* transgene in the MBs (*rut²⁰⁸⁰; LexAop-spH/UAS-rut; TH-LexA/30Y-GAL4*) upon AL + NMDA stimulation. N = 8. (**D**) spH responses upon AL + NMDA stimulation in indicated lines. Two-way ANOVA indicates significant differences due to time point ($F_{3,84}$ = 6.69, p<0.0164) and genotype ($F_{2,84}$ = 34.70, p<0.0001). N = 9 for all genotypes.

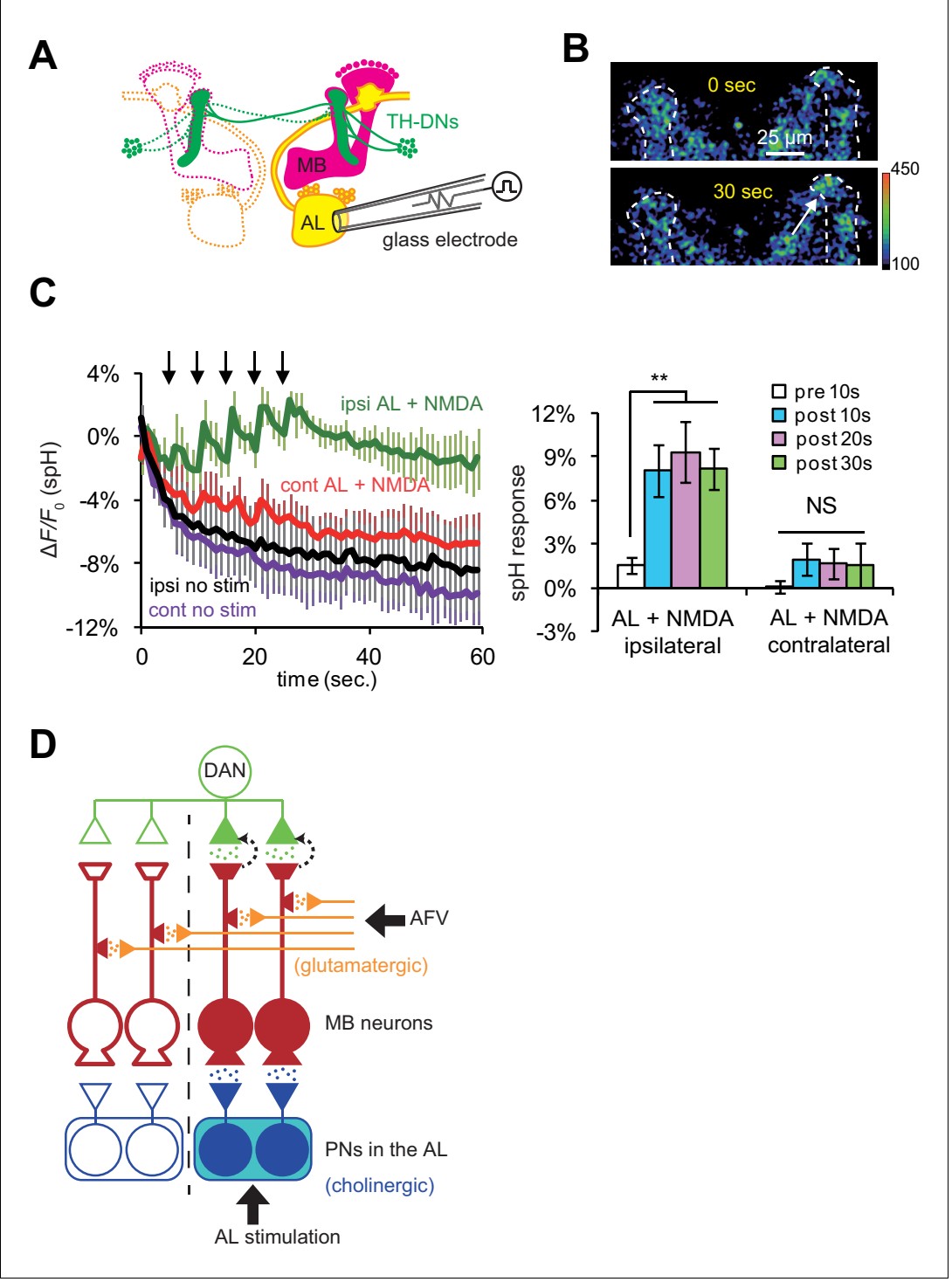

**Figure 7.** DA release is restricted to coincidentally postsynaptic MB neurons. (**A**) Schematic diagram of the connections between ALs, MBs, and TH-DA neurons. PNs (yellow) connect ALs only to ipsilateral MB calyces, whereas DA neurons (green) innervate MBs bilaterally. (**B**) Typical fluorescence images of spH at TH-DA neuron terminals on the vertical lobes 0 s and 30 s after AL + NMDA stimulation. The arrow indicates the vertical lobe of the MB (dotted lines) ipsilateral to the stimulated AL. (**C**) Left panel, changes in spH fluorescence at TH-DA terminals on ipsilateral and contralateral MB lobes receiving coincident AL + NMDA stimulation. N = 5. Right panel, one-way ANOVA indicates significant differences in spH fluorescence in the ipsilateral MB ($F_{3,16}$ = 5.333, p=0.001), but not the contralateral MB ($F_{3,16}$ = 0.458, p=0.716). N = 5. (**D**) Proposed model of a cellular circuit regulating DA release and LTE formation. Simultaneous activation of cholinergic AL inputs (blue) and

*Figure 7 continued on next page*

*Figure 7 continued*

glutamatergic AFV inputs (orange) to MB neurons activates a gating mechanism that allows DA (green dots) to be released onto these coincidently activated MB neurons. DA is not released onto MB neurons that only receive single inputs. PNs and ALs are outlined in blue, MB neurons are outlined in red, inputs from the AFV are shown in orange, and a DA neuron is shown in green. Released glutamate from AFV inputs and acetylcholine from PNs are shown by orange dots and blue dots, respectively. Filled areas represent activated (depolarized) regions.

we propose that it is likely related to learning and memory-associated plastic changes. Aversive olfactory learning involves associations between odor information, which is transmitted to the MBs via PNs from the antennal lobes, and shock information, which is likely transmitted to the brain via the AFV. LTE is induced by simultaneous stimulation of the same sensory inputs, and shares various physiological phenotypes and molecular requirements with learning. In the current manuscript, we further demonstrate that both LTE and learning (3 min memory) require the activity of NRs in the MBs, and identify a subset of glutamatergic neurons (using the GMR51C07 line), which are required for memory, and exhibit vesicular exocytosis onto MB vertical lobes upon AFV stimulation.

Our synapto-pHluorin data indicate that AL stimulation evokes vesicular exocytosis from cholinergic PN terminals in the MB calyx, a result consistent with previous studies (*Heisenberg, 2003*; *Su and O'Dowd, 2003*; *Yasuyama et al., 2002*). However, we find that AFV stimulation evokes exocytosis from glutamatergic neurons rather than DANs. Significantly, direct application of NMDA to the MBs can replace AFV stimulation during LTE induction. NMDA acts through NRs in the MBs to induce LTE. Thus, although the complete neural pathway from the AFV to the MBs has not yet been identified, our results suggest that the final step in transmission of US information from the AFV to the MBs occurs through glutamatergic neurons activating MB NRs.

DA release occurs at a step after cholinergic and glutamatergic activation of the MBs, since LTE induced by AL + NMDA stimulation requires D1Rs, while LTE induced by the DA application does not require NR activity or AL stimulation. Furthermore, DA application induces LTE even in the presence of TTX in $Ca^{2+}$ free media, suggesting that DA release and activation of D1Rs in the MBs comprise the last intercellular step during LTE establishment. Similar to DA-induced plasticity in AL-MB transmission, previous papers have also found that exogenous activation of dopaminergic pathways can induce plasticity in MB Kenyon cell to MB output neuron transmission in *Drosophila* (*Cohn et al., 2015*), and in CA3 to CA1 transmission in rats and guinea pigs (*Huang and Kandel, 1995*; *Williams et al., 2006*).

DA release requires the activity of the Rut-AC in the MBs. Rut-AC is proposed to function as a coincidence detector of olfactory and somatosensory inputs during olfactory learning (*Boto et al., 2014*; *Busto et al., 2010*; *Heisenberg, 2003*; *Tomchik and Davis, 2009*), suggesting that coincident AL and AFV-mediated activation of postsynaptic Kenyon cells functions to gate presynaptic dopaminergic release through a Rut-AC-dependent mechanism. Thus, simultaneous AL and AFV inputs specify the appropriate targets where DA needs to be released, while the amount of released DA may regulate the occurrence and intensity of plastic changes.

While our data demonstrate that Rut-AC functions at a step in between coincident activation of Kenyon cells and presynaptic DA release, we note that we have not proven that Rut-AC actually detects coincident AL and AFV activity. Previous studies have shown that Rut-AC is activated by both $Ca^{2+}$/calmodulin and G-protein-mediated signaling, and may function to integrate these two signaling pathways (*Levin et al., 1992*; *Tomchik and Davis, 2009*). However, nAChRs and NRs are both ionotrophic receptors, and it is unclear whether or how either receptor activates G-proteins. Alternatively, Tsien and colleagues have recently demonstrated that the activation of neuronal gene expression by L-type voltage gated $Ca^{2+}$ channels requires two independent activation mechanisms: $Ca^{2+}$ influx, and a separate voltage-dependent conformational change (*Li et al., 2016*). Neither signal alone is sufficient for activation. Similarly, Rut-AC, or another coincidence detector in the MBs may require convergent activation by $Ca^{2+}$ influx from NRs, and a conformational change induced by action potentials propagated from the calyx.

Our current study focuses on the identification of a novel DA gating mechanism involved in plasticity in general. Thus, at this point, we have not examined the precise relationship between this gating mechanism and a specific phase of memory. In *Drosophila*, formation and storage of short-term

memories (STMs) require γ and α′/β′ Kenyon cells, while consolidation to long-term memories (LTMs) is proposed to require α/β Kenyon cells (*Guven-Ozkan and Davis, 2014*; *Krashes et al., 2007*). In our study, we measured LTE at the vertical lobes, and previously demonstrated that LTE occurs in both the α and α′ lobes (*Ueno et al., 2013*). Thus LTE may be associated with both STM and LTM. Consistent with a role of LTE in LTM, a previous study by Blum et al. reported that *rutabaga* expression in α/β neurons is necessary for LTM, but not STM (*Blum et al., 2009*). Since we examined LTE, in part in the α lobes, and demonstrate that it requires *rutabaga*, the LTE we examined may be necessary for LTM formation. On the other hand, we also show that NR expression in the MBs is required for both learning (3 min memory) and LTE. Although NRs are also required LTM, this requirement is in the ellipsoid bodies, a brain region distinct from the MBs (*Wu et al., 2007*). Taken together, our data suggest that LTE may be important for both STM and LTM, and may be a general mechanism for DA-mediated plasticity.

Finally, we note that our results show an apparent discrepancy with previous work demonstrating that artificial activation of DANs can replace US exposure during learning (*Aso et al., 2010*; *Burke et al., 2012*; *Claridge-Chang et al., 2009*; *Liu et al., 2012*). Our results suggest that DA on its own is sufficient to induce LTE, and coincident activation of odor-specific and US-specific neurons is required to specify unique locations where DA release is gated. Thus, if release is artificially induced in a large subset of DANs, one would expect that odor-specific learning would not occur, and instead flies would display a general avoidance or attraction to odors. While we cannot explain why this does not happen, we note that artificial activation of DANs in other studies is induced using temperature or optically regulated cationic (TrpA1 or $P2X_2$) channels (*Aso et al., 2010*; *Burke et al., 2012*; *Claridge-Chang et al., 2009*; *Liu et al., 2012*). We propose that the activation of these cationic channels by itself is not sufficient to induce dopaminergic release since our proposed post-synaptic gating mechanism is not induced in this situation. However, coincident application of an odor and heat or intensive light may be sufficient to open this gate. In combination with TrpA1 or $P2X_2$-dependent activation of DANs, this is sufficient to generate odor-specific memories.

Our data suggest that LTE formation follows a circuitous route. Coincident signals are first detected in the MBs, but formation of LTE requires communication to external DA neurons, which then transmit a signal back to MBs. Why is such an apparently non-parsimonious pathway used? The basic input requirements for forming associations are two temporally paired stimuli. However, it is clear from both vertebrate and invertebrate systems that the strength of resulting associations is modulated by various factors, including the physiological state of the organism. While some physiological states may alter how the organism perceives the incoming sensory information, other states may modulate the probability of forming an association independently of sensory perception. Incorporating a dopaminergic feedback loop may allow the organism to fine tune the amount of learning that occurs based factors including the physiological state of the organism.

## Materials and methods

### Fly stocks

All stocks were raised on standard cornmeal medium at 25°C ± 2°C and 60% ± 10% humidity under a 12/12 hr light–dark cycle. Flies were used for experiments 1–3 d after eclosion.

### Transgenic lines

*UAS-G-CaMP* (*Reiff et al., 2005*), *UAS-G-CaMP3* (RRID:BDSC_32234, Bloomington Stock Center, Indiana) and *LexAop-G-CaMP2* (*Ueno et al., 2013*) lines were used for measuring $Ca^{2+}$ responses, *UAS-synapto-pHluorin* (*UAS-spH*) (*Ng et al., 2002*) was used for measuring vesicle release, *P[TRiP. JF02044]attP2* (RRID:BDSC_26019, Bloomington Stock Center) was used for *Nmdar2* RNAi, *MB-LexA::GAD* (*Ueno et al., 2013*) was used for the MB *LexA* driver, and *OK107*, *c309* and *c747* (*Aso et al., 2009*) were used for MB *GAL4* drivers. *TH-GAL4* (*Friggi-Grelin et al., 2003*) and *TH-LexAp65* (provided by Y. Aso and J. Rubin, Janelia Research Campus, Ashburn, Virginia, USA) were used for TH-DA neuron drivers, *GH146-GAL4* (*Stocker et al., 1997*) (RRID:BDSC_30026) was used for the PN driver, *P[EP]Nmdar1*^EP331^ (*Xia et al., 2005*) (RRID:BDSC_17112) was used for the *Nmdar1* mutant, and *GMR51C07-GAL4* (*Jenett et al., 2012*) (RRID:BDSC_38773, Bloomington Stock Center,), which carries approximately 3 KB of the 5′ flanking sequence of the *VGlut* gene was used for

the *VGlut-GAL4* driver. *UAS-nAChRα7*$^{Y195T}$ (*Mejia et al., 2013*), containing a dominant negative form of the *nAChRα* subunit of nAChRs, was used to disrupt nAChR function. *LexAop-R-GECO1* was provided by A. Nose, Tokyo University, Tokyo, Japan.

To express spH in *TH-LexA* labeled DA neurons, we generated *LexAop-spH* flies. spH cDNA (*Ng et al., 2002*) was amplified by PCR and subcloned into pCasper-lexAop-W (*Diegelmann et al., 2008*). The resulting construct was injected into embryos to obtain *LexAop-spH* flies (Genetics Service Inc., USA).

## Isolated whole brain preparation

Brains were prepared for imaging as previously described (*Ueno et al., 2013*). Briefly, brains were dissected in ice cold $Ca^{2+}$ free HL3 medium (in mM, NaCl, 70; sucrose, 115; KCl, 5; MgCl$_2$, 20; NaHCO$_3$, 10; trehalose, 5; Hepes, 5; pH 7.3) (*Stewart et al., 1994*), and placed in a recording chamber filled with normal, room temperature HL3 medium (the same recipe as above, containing 1.8 mM CaCl$_2$). To deliver DA through the blood brain barrier, brains were treated with papain (10 U/ml) for 15 min at room temperature, and washed several times with $Ca^{2+}$ free HL3 medium prior to use (*Gu and O'Dowd, 2007*).

## Imaging analysis

Fluorescent images were captured at 15 Hz using a confocal microscope system (A1R, Nikon Corp., Tokyo, Japan) equipped with a 20x water-immersion lens (numerical aperture 0.5; Nikon Corp). To measure AL- and AFV-induced $Ca^{2+}$ responses, ALs and AFVs were stimulated (30 pulses, 100 Hz, 1.0 ms pulse duration) using glass micro-electrodes. Fluorescent traces from 3 stimulus trains with an inter-train interval of 10 s were averaged to obtain $F$ values, and the 5 sequential frames before stimulus onset were averaged to obtain $F_0$ values. These were used to calculate $\Delta F/F_0$. To quantify LTE, we divided peak $\Delta F/F_0$ values, obtained from AL-induced $Ca^{2+}$ responses at indicated times after LTE induction, by peak values obtained before LTE induction. To induce LTE by AL + AFV stimulation, both the AL and AFV were simultaneously stimulated with twelve stimulus trains (inter-train interval of 5 s) (*Ueno et al., 2013*). For inducing LTE by AL stimulation and puff application of NMDA, 1 mM NMDA (in HL3 medium containing 4 mM $Mg^{2+}$) was applied from a micropipette onto the MBs for 1 min (pressure = 6 psi) using a Picospritzer III system (Parker Hannifin Corp., USA). During the second minute of application, ALs were stimulated with 6 stimulus trains (inter-train interval of 5 s). To induce LTE by AL stimulation and bath application of 100 μM NMDA, NMDA was diluted in HL3 containing 4 mM $Mg^{2+}$, to prevent $Mg^{2+}$ block (*Miyashita et al., 2012*), and added to the recording chamber while a peristaltic pump (15 ml/min) was used to remove excess fluid from the chamber. For washout, media was removed from the recording chamber and continuously replaced with fresh HL3 buffer (20 mM MgCl$_2$). ALs were stimulated with 5 or 12 stimulus trains (with an inter-train interval of 5 s) during one min NMDA addition. To induce LTE by DA application alone, DA was added to the recording chamber for 1 min in the absence of AL or AFV stimulation and subsequently washed out. Receptor antagonists were added and washed out similarly. $\Delta F/F_0$ for spH was measured similarly to measurements of $\Delta F/F_0$ for $Ca^{2+}$ responses. To quantify stimulation-induced fluorescent changes of spH, $\Delta F/F_0$ in the absence of stimulation was subtracted from $\Delta F/F_0$ at indicated time points during and after stimulation.

## Chemicals

NMDA, butaclamol ((+)-butaclamol hydrochloride), and SCH23390 (R(+)-SCH-23390) were purchased from Sigma-Aldrich (Missouri, USA). DA (3,4-dihydroxyphenethylamine hydrochloride) was purchased from Wako Pure Chemical Industries (Osaka, Japan). Papain was purchased from Worthington Biochemical Corporation (New Jersey, USA). Butaclamol was dissolved in DMSO, and NMDA, SCH23390 and DA were dissolved in water.

## Statistics

All data in bar and line graphs plot means ± SEMs. Student's t-test was used to evaluate the statistical significance between two data sets. For multiple comparisons, one-way or two-way ANOVA was used, followed by Bonferroni post hoc analyses using Prism software (GraphPad Software, Inc., La

Jolla, CA, USA) or XLtoolbox (Free Software Foundation, Inc. MA, USA). In all figures, NS indicates p>0.05, * indicates p<0.05, and ** indicates p<0.01.

## Acknowledgements

We thank A Nose for *UAS-R-GECO1* transgenic flies, Y Aso and J Rubin for *TH-LexAp65* transgenic flies, and G Miesenbock for spH cDNA. We also thank T Miyashita, M Matsuno, T Ueno and Y Hirano for helpful discussions.

## Additional information

### Funding

| Funder | Grant reference number | Author |
|---|---|---|
| Japan Society for the Promotion of Science | JP21700376 | Kohei Ueno |
| Ministry of Education, Culture, Sports, Science, and Technology | Memory Dynamism, JP25115006 | Minoru Saitoe |
| Takeda Science Foundation | | Minoru Saitoe |
| Japan Society for the Promotion of Science | JP15K06729 | Junjiro Horiuchi |
| Japan Society for the Promotion of Science | JP15K14331 | Ema Suzuki |
| Japan Society for the Promotion of Science | JP15K18577 | Shintaro Naganos |

The authors declare that there was no funding for this work.

### Author contributions

KU, Conceptualization, Data curation, Software, Formal analysis, Validation, Investigation, Methodology, Writing—original draft; ES, Resources, Data curation, Validation, Investigation; SN, Data curation, Validation, Investigation; KO, Resources, Investigation, Project administration; JH, Conceptualization, Data curation, Validation, Writing—original draft, Writing—review and editing; MS, Conceptualization, Data curation, Formal analysis, Supervision, Funding acquisition, Validation, Investigation, Writing—original draft, Project administration, Writing—review and editing

### Author ORCIDs

Minoru Saitoe, http://orcid.org/0000-0001-9731-4214

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
