## [Decision Letter]

Thank you for submitting your article "Coincident postsynaptic activity gates presynaptic dopamine release to induce plasticity in *Drosophila* mushroom bodies" for consideration by *eLife*. Your article has been favorably evaluated by Eve Marder as the Senior Editor and three reviewers, one of whom, Leslie C Griffith (Reviewer #1), is a member of our Board of Reviewing Editors. The following individuals involved in review of your submission have agreed to reveal their identity: Josh Dubnau (Reviewer #2) and Yi Zhong (Reviewer #3).

The reviewers have discussed the reviews with one another and the Reviewing Editor has drafted this decision to help you prepare a revised submission.

Summary:

This paper addressed the organization and plasticity of the MB memory circuit. This is a venerable and well-studied circuit, but the actual cellular processes underlying associative memory formation are not completely understood. This paper proposes a model for long-term enhancement (LTE) of AL-driven MB responses, a purely electrophysiological phenomenon, which posits glutamatergic/MB activity gating of the US dopamine signal and provides a mechanism for ensuring DA signaling is local. The idea is well-supported by pharmacological and genetic evidence using imaging and electrophysiological methods. The one weak point of the paper is that there is no behavioral verification of the model to firmly link the LTE phenomenon to actual memory, and in fact there is a paper in the literature that suggests MB NMDARs are *not* essential for LTM (Wu et al., 2007). Without behavioral results, it remains possible that LTE is an epiphenomenon and not part of the underpinnings of memory.

Essential revisions:

All the reviewers found this study very interesting and potentially important. But the lack of evidence linking LTE directly to LTM is concerning. The authors should add some discussion of the links to behavior and specifically discuss the Wu et al. paper.

---

## [Author Response]

*Essential revisions:*

*All the reviewers found this study very interesting and potentially important. But the lack of evidence linking LTE directly to LTM is concerning. The authors should add some discussion of the links to behavior and specifically discuss the Wu et al. paper.*

1) Strengthen/solidify the connection between LTE and learning and memory.

We have added two behavioral experiments to strengthen the connection between LTE and learning/memory.

First, we demonstrate that inhibiting activity of *GMR51C07* glutamatergic neurons reduces 15 min memory. In our in vitro LTE system, we show that AFV stimulation induces vesicular release from *GMR51C07* glutamatergic neurons onto the MB vertical lobes. If this release is important for transmitting electrical shock information to the MBs, inhibiting *GMR51C07* activity in vivo using *shi^ts^* should cause a learning or memory phenotype. We find this to be the case. While memory is normal when flies are trained and tested at the permissive temperature, memory is significantly reduced when they are trained at the restrictive temperature, and tested at the permissive temperature. We have added this data as Figure 1—figure supplement 1.

Second, we demonstrate that NMDA-type glutamate receptors (NRs) in the MBs are required for normal learning. In our manuscript, we present evidence indicating that somatosensory information is conveyed to the MBs through glutamate acting on NRs. NRs in the MBs are required for LTE, and application of NMDA can bypass the requirements for AFV stimulation. If LTE is linked to learning, inhibition of NRs in the MBs should cause learning defects. In our revision, we added an experiment showing that knockdown of NRs in the MBs inhibits 3 min memory. This data has been added as Figure 2—figure supplement 1.

In addition we have added extensive discussion to the Introduction and Discussion sections describing the molecular and physiological similarities between learning and LTE.

2) Discuss apparent contradictions between our results and those of Wu et al. (2007) who demonstrate that LTM does not require NRs in the MBs.

While we do not know precisely which phase of learning and memory LTE is associated with, we have evidence suggesting that it may be important for both short and long-term memories. Thus, even though Wu et al. demonstrate that MB NRs are not required for LTM, this is not necessarily inconsistent with our LTE data since results from our lab as well as others (Xia et al., 2005; Miyashita et al., 2012; and Figure 2—figure supplement 1 in our current manuscript), suggest that MB NRs are required for learning. We have added the Wu et al. reference to our revised manuscript and address this issue in the sixth paragraph of the Discussion section.